# Spinal Deformities in Wild Reptiles: A Systematic Review and Meta-Analysis

**DOI:** 10.3390/biology14091119

**Published:** 2025-08-24

**Authors:** Gergely Horváth

**Affiliations:** 1Department of Systematic Zoology and Ecology, Institute of Biology, ELTE Eötvös Loránd University, Pázmány Péter Sétány 1/c, 1117 Budapest, Hungary; gergely.horvath@ttk.elte.hu; 2HUN-REN–ELTE–MTM Integrative Ecology Research Group, Pázmány Péter Sétány 1/c, 1117 Budapest, Hungary

**Keywords:** spinal malformations, kyphosis, scoliosis, reptiles, meta-analysis

## Abstract

Spinal deformities, such as twisted, curved, or humped backs, are seen in many animals, including reptiles, but scientists still know little about how common these problems are in wild reptiles or what causes them. This study gathered the most complete set of records on spinal deformities in wild reptiles, using published research, online reports, and social media inquiries. In total, 690 reports were collected, involving 109 species from around the world, with turtles and tortoises making up most of the cases. The general frequency of spine deformities was found to be 0.21%, though this may be an underestimate. Although the frequency of deformities is not linked to evolutionary differences, habitat use, or habitat type, some patterns were found. For example, curved backs were more common in aquatic turtles, while sideways curvatures were more often seen in land-dwelling lizards and snakes. While these deformities could affect survival, growth, or reproduction, there are few solid data on their real-life impacts. The study shows that these spine problems are widespread and likely under-reported, and this calls for future research to use more detailed and consistent methods. Understanding these deformities better could help scientists and conservationists protect wild reptiles and their environments.

## 1. Introduction

Spinal column deformities are observed in every major vertebrate group, with reported cases in fish [1,2], amphibians [3,4], birds [5], and mammals [6,7]. Unsurprisingly, reptiles are no exception. In fact, the oldest known record of spinal deformities comes from a Permian aquatic parareptile, *Stereosternum tumidum*, exhibiting scoliosis (lateral deviation of the vertebral column) [8]. Other common forms of these deformities include kyphosis (abnormal increase of the posterior convexity of the vertebral column) and lordosis (abnormal concave curvature of the vertebral column). Deformities can also occur in combination, with the most common being kyphoscoliosis (Figure 1), while the simultaneous presence of all three conditions is referred to as rhoecosis [9,10].

The earliest scientific reports of spinal deformities in reptiles date back to the early 1900s, with all initial cases involving Testudines. Wandolleck [11] offers a detailed examination of a kyphotic Hermann’s tortoise (*Testudo hermanni*) from the Dresden Zoo, though he refrains from speculating on the exact cause of the deformity. Similarly, Vogt [12], Gressit [13,14,15], Mertens [16], and Necker [17] describe their observations of ‘humpbacked’ specimens as curiosities. Although some of these authors mention factors such as environmental contamination and growth disorders as potential causes of the deformities, they fail to provide a broader context for the pathology or discuss the potential impact of the condition on the affected individuals. While the aetiology and pathogenesis of these conditions still remain unclear, spinal deformities appear to be highly multifactorial, with no single aetiopathological cause that can be fully accounted for. Importantly, our understanding of the causes and development of spinal deformities is largely based on human medicine, but due to the anatomical similarities of the spine across vertebrates, animal models are frequently studied [18,19,20]. However, despite structural similarities, certain forms of spinal deformities seem to be taxon specific. For example, lumbar lordosis and idiopathic scoliosis are conditions predominantly associated with humans, likely as a consequence of bipedal posture [21,22]. Additionally, the factors contributing to spinal deformities, as well as the mechanisms of their development, may vary across different taxa.

### 1.1. The Most Common Factors Behind Spinal Deformities in Reptiles

Although physical trauma and vertebral fractures were previously suggested as primary causes of spinal deformities and such injuries can indeed lead to kyphosis or scoliosis, occurrences are rather sporadic or secondary [9,23]. In contrast, pathological fractures resulting from nutritional or metabolic diseases are more likely to cause spinal deformities (see below). Perhaps the most diverse group of aetiological agents for kyphosis and scoliosis has been proposed in the case of Testudines. Early authors suggested that these deformities emerge due to differential growth rates of the skeletal components of the carapace, possibly in combination with the premature fusion of the vertebral column with the carapacial plates [24]. Based on observations of hatchlings, Cagle [25] and Williams [26] proposed that abnormal yolk retraction could lead to early skeletal fusion of the carapace and, consequently, a malformed spine. However, this idea was not further tested, although it has been frequently referenced. Another possibility is that spinal deformities have a neoplastic origin [27,28], though cases that clearly attributable for abnormal tumour growth have not been documented. As in other vertebrates, spinal deformities in reptiles are typically multifactorial abnormalities and often represent just one clinical sign within a complex disease profile. Given the relative prevalence of spinal deformities in captive reptiles, advancements in reptile veterinary medicine have allowed for the identification of key contributing factors [29].

#### 1.1.1. Anomalies of Embryonic Development

In reptiles, kyphosis, scoliosis, lordosis, and their combinations are commonly reported congenital malformations. A study by Sant’Anna et al. [30] found that spinal deformities were the most frequent malformations in newborn jararacas (*Bothrops jararaca*) and Cascadel rattlesnakes (*Crotalus durissus*). Although the overall prevalence of all malformations was negligible (2.3%), among malformed neonates, kyphosis alone was accounted for more than 60% of the cases in both species (*B. jararaca*: 67.4%; *C. durissus*: 75%). In contrast, kyphosis was less common in sea turtles; for example, in the olive ridley sea turtle (*Lepidochelys olivacea*), kyphosis was present in only 13.2% of malformed embryos or neonates [31], with some studies reporting even lower prevalence rates [32,33]. Importantly, although spinal deformities often appear as standalone conditions, in many cases, they are combined with other deformations, e.g., cranial malformations, anophthalmia, buphthalmia, coiled tail, and schistogastria [30,34,35,36]. These multifactorial defects are not necessarily fatal, and if they do not interfere with locomotion and feeding, individuals might reach adulthood [37].

Identifying the primary causes of congenital spinal deformities is challenging due to the intricacy of vertebrate embryonic development. This highly synchronised biological process is vulnerable to aberrations and errors at various stages including gametogenesis, fertilisation, blastogenesis, embryogenesis, and fetogenesis [10]. While many congenital spinal deformities have a genetic origin, suboptimal environmental conditions also play a significant role. However, the interaction between genetic and environmental influences remains poorly understood, making it difficult to attribute developmental anomalies to a single cause. Environmental stressors may also influence the epigenetic regulation of gene expression, either directly or indirectly through downstream pathways. Although the role of epigenetic regulation in the embryonic development of spinal deformities is not yet conclusive, recent studies on the loggerhead sea turtle (*Caretta caretta*) suggest that epigenetic mechanisms may contribute to reduced fitness and sublethal shell abnormalities [38,39]. Regarding genetic underpinnings, geographic isolation and population fragmentation are perhaps the most significant factors contributing to the restriction of the genetic pool in natural populations. Inbreeding and genetic drift, along with the subsequent expression of genes associated with developmental anomalies, often result from this restriction. Thus, in small and isolated populations, the frequency of skeletal deformities is found to be increased compared to those with high genetic variability [40,41,42].

The influence of agents originating from both the external and maternal environments can significantly impact embryonic development in both oviparous and viviparous reptiles, as they both employ a simple form of placentation [43]. Embryos of several oviparous reptile taxa are vulnerable to extremes in moisture, gas exchange, and incubation temperature. Insufficient environmental humidity is frequently cited and investigated as a key teratogen in abnormal shell development in Testudines, often leading to spinal deformities. Lynn and Ullrich [44] found that suboptimal soil moisture during egg incubation can produce various abnormalities of the carapace of freshwater turtles, such as the painted turtle (*Chrysemys picta*) and the common snapping turtle (*Chelydra serpentina*). Regarding extreme temperatures, Idrisova [45] found that both grass snakes (*Natrix natrix*) and sand lizards (*Lacerta agilis*) incubated at high temperatures (29 °C and 34 °C) more frequently exhibited a wide range of deviations, including spinal deformations. Additionally, anomalies were more pronounced in these hatchlings, resulting in reduced survival. Similarly, Telemeco et al. [46] reported that prolonged exposure to extreme incubation temperature increases the frequency of shell and scute abnormalities in painted turtles (*C. picta*). This is significant, as in Testudines, abnormal shell development can often lead to kyphosis [27,47].

Perhaps the most important emerging group of environmental teratogens for reptiles are pollutants [48,49,50]. Since the 1990s, research on environmental toxicology in reptiles has increased, though it still receives less attention compared to contaminants in other vertebrates. Nonetheless, it is now clear that anthropogenic activities are seriously affecting natural reptile populations globally through various chemical substances, including heavy metals, pesticides, and certain drugs. Considerable data suggest that in contaminated and highly urbanised areas, the incidence of congenital abnormalities is significantly higher [34,51,52,53].

#### 1.1.2. Nutritional and Metabolic Diseases

In captivity, the most common cause of skeletal spinal abnormalities is metabolic bone diseases (MBDs). MBD is not a single condition but rather an umbrella term that encompasses a group of related conditions that affect the integrity and function of bones [47,54,55]. The most common MBD is nutritional secondary hyperparathyroidism (NSHP), although renal secondary hyperparathyroidism can also occur. NSHP typically arises from inadequate UV-B radiation, an improper temperature range, or insufficient dietary calcium or vitamin D_3_ [54,55,56]. While NSHP can potentially affect all reptile taxa, it is more prevalent in heliothermic lizards and Testudines. These reptiles, due to their mostly insectivorous and herbivorous diets, cannot rely solely on dietary sources of vitamin D_3_; thus, the activation of dietary provitamin D precursors via UV-B radiation is essential. Typical clinical signs of MBDs include pathological fractures, curvature of the spine, or deformed carapace in Testudines, and many bones appearing more curved than normal [47,55,57].

It is noteworthy to mention a condition affecting Testudines known as shell pyramiding [47]. Pyramiding refers to the abnormal growth of the scutes and the underlying bones of the carapace and is considered a distinct form of nutritional and metabolic disease. In affected specimens, material deposition between the scutes accelerates scute growth at a rate faster than the expansion of the underlying bone. This forces the keratin and bone to grow conically upward, causing a separation between the vertebrae and the carapace, with only extremely thin bone bridging from the dorsal vertebral body to the carapace. This condition commonly results in kyphosis, scoliosis, and lordosis. Pyramiding is thought to be a multifactorial process, likely caused by an imbalanced calcium-to-phosphorus ratio, increased growth rates, and low humidity [58,59].

In contrast to lizards and Testudines, snakes appear to be less affected by MBDs because they consume their prey whole, providing sufficient dietary vitamin D_3_ and calcium. Additionally, while MBD and shell pyramiding are very common in captivity, these conditions are believed not to affect wild populations [47,56]. However, their potential presence in natural populations cannot be entirely ruled out.

#### 1.1.3. Infection-Induced Osteopathy

Bacterial osteomyelitis and osteoarthritis, both being severe bone infection, are common in reptiles and can lead to stiffness, pathological fractures, and spinal deformities when the spinal column is affected [60]. Vertebral osteomyelitis and osteoarthritis are especially common in snakes [61,62,63]. The bacteria most commonly associated with this condition are Gram-negative rods including *Citrobacter* sp., *Salmonella* sp., *Proteus* sp., *Pseudomonas* sp., and *Escherichia coli* [62,64]. Gram-positive bacteria seem to rarely cause osteomyelitis in reptiles; however, there are documented cases involving *Staphylococcus* sp., *Streptococcus* sp., and *Enterococcus* sp. [61,63]. On rare occasions, fungi have been reported as causative agents of bone infection [65].

Bacteria often invade the spinal column through direct extension, typically resulting from wound contamination or trauma to nearby soft tissues. In the case of *Salmonella* sp., a common inhabitant of the reptile digestive tract, bacterial spread can occur during periods of immune suppression. This may lead to septicaemia, allowing the bacteria to reach the spinal column via hematogenous dissemination [66]. Some researchers have proposed that osteomyelitic spinal lesions could be caused by bacterial endotoxins [61]. For a definitive histological diagnosis of the bacterial origin of osteomyelitis and osteoarthritis, isolating bacteria from bone cultures is essential. This requirement may explain why reports in the literature almost exclusively involve captive animals. However, Isaza et al. [61] observed a strong correlation between bone culture and blood culture results, suggesting that blood cultures, being a simpler and far less invasive method, could be a feasible alternative for diagnosing active bacterial osteomyelitis and osteoarthritis.

### 1.2. Aims and Goals

The number of published cases and the diversity of affected species have notably increased since the 2000s, likely due to growing scientific interest in ‘humpbacked’ reptiles. Despite this, spinal deformities remain relatively rare in wild populations. This rarity may be attributed to higher mortality rates among affected individuals [67]. However, several reports describe adult reptiles with spinal deformities that appear unaffected in terms of foraging and mobility [67,68,69,70,71], suggesting that such deformities may not necessarily impact survival. Mitchell and Johnston [72] also found that the growth patterns of a Florida chicken turtle (*Deirochelys reticularia chrysea*) with kyphoscoliosis were comparable to those of unaffected individuals in the same population.

Conversely, it has been suggested that increased environmental stressors, such as human-induced changes and pollution, may be contributing to the rising frequency of spinal deformities. While no definitive link has been established between spinal deformities and contaminants, the widespread presence of pesticides, herbicides, fungicides, and polycyclic aromatic hydrocarbons (PAHs) poses potential risks even to (quasi-)natural populations [73]. In line with this, several observations have linked agrochemicals to spinal malformations [69,74].

It also remains unclear whether different habitat use strategies—terrestrial, arboreal, aquatic, or semiaquatic—make reptiles more prone to spinal deformities or whether certain deformities are more common in specific habitats. For instance, kyphosis is thought to primarily affect aquatic species (e.g., turtles and crocodiles) [75,76], while scoliosis appears more frequent in terrestrial species (e.g., snakes and lizards) [77,78,79].

The primary objective of this study is to provide the first comprehensive review of spinal deformities in wild reptiles, incorporating both published reports and previously unpublished personal observations. The review focuses on the occurrence of various spinal deformities across reptile families, with prevalence data extracted or calculated where possible. These data were analysed using a phylogenetic meta-analysis to estimate a global effect size for spinal deformity frequency and evaluate whether habitat type (natural, seminatural, or urbanised) or habitat use strategy (aquatic, semiaquatic, terrestrial, or arboreal) influences prevalence. Based on the findings, I aim to discuss the potential ecological and behavioural effects of spinal malformations and provide recommendations for future research.

## 2. Methodology

### 2.1. Data Search

This review conforms to the PRISMA (Preferred Reporting Items for Systematic Reviews and Meta-Analyses) guidelines [80]. The PRISMA checklist can be found in the Electronic Appendix A. I compiled records on spinal deformities in reptiles from the published, peer-reviewed literature, as well as the non-peer reviewed (‘grey’) literature, spanning the earliest available material (1904) to January 2025. Given that most of the records in this systematic review consist of single-observation natural history notes, short communications, or research articles published in society journals that are not indexed by major databases such as *Scimago* or *Scopus*, a traditional Boolean query was not applicable. Instead, I systematically reviewed the relevant sections of the complete published issues of the following journals: *Boletín de la Associacón Herpetológica Española* https://herpetologica.es/category/publicaciones/boletin-de-la-asociacion-herpetologica/ (accessed on 6 February 2025) (1990–2024), *Bulletin of the Chicago Herpetological Society* https://chicagoherp.org/bulletins-2 (accessed on 6 February 2025) (1990–2023), *Herpetology Notes* https://herpetologynotes.org/index.php/hn/issue/archive (accessed on 6 February 2025) (2014–2025), *Herpetological Review* https://ssarherps.org/herpetological-review-pdfs/ (accessed on 6 February 2025) (1967–2024), *Herpetozoa* https://herpetozoa.pensoft.net/issues (accessed on 6 February 2025) (2019–2025), *The Herpetological Bulletin* https://www.thebhs.org/publications/the-herpetological-bulletin (accessed on 6 February 2025) (1980–2024), *The Herpetological Journal* https://www.thebhs.org/publications/the-herpetological-journal (accessed on 6 February 2025) (1985–2025), *Mesoamerican Herpetology* http://www.mesoamericanherpetology.com (accessed on 6 February 2025) (2014–2017), *Salamandra* https://www.salamandra-journal.com/index.php (accessed on 6 February 2025) (1965–2024), *Reptiles & Amphibians* https://journals.ku.edu/reptilesandamphibians/issue/archive (accessed on 6 February 2025) (2009–2025), *Revista Española de Herpetología* https://herpetologica.es/category/publicaciones/revista-espanola-de-herpetologia/ (accessed on 6 February 2025) (1996–2010), *Revista Lationoamericana de Herpetología* https://herpetologia.fciencias.unam.mx/index.php/revista/issue/archive (accessed on 6 February 2025) (2018–2024), and *Russian Journal of Herpetology* http://rjh.folium.ru/index.php/rjh/issue/archive (accessed on 6 February 2025) (1994–2024). These journals often contain relevant but niche material, including observations that may not be indexed in standard scientific databases.

In addition to reviewing these journal issues, I consulted various online libraries and databases such as *Google Scholar* https://scholar.google.com/ (accessed on 13 February 2025) and *Web of Science* https://www.webofscience.com/wos (accessed on 13 February 2025), where I searched for relevant terms like ‘kypho’, ‘scolio’, ‘lordo’, ‘spine’, ‘spinal’, ‘deform’, and ‘malform’. I also conducted searches in multiple languages (German, Spanish, Portuguese, and French) to ensure comprehensive coverage. To identify photographic records, I searched visual platforms such as *Flickr* https://www.flickr.com/ (accessed on 10 March 2025) and *Google Images*
https://images.google.com/ (accessed on 10 March 2025). I also retrieved and reviewed any records cited in short notes as containing relevant information. For materials that were not available online, including physical journals and books, I made requests to libraries or directly contacted individual researchers. Lastly, unpublished observations were solicited by circulating requests in several online herpetological groups (*California Field Herping* https://www.facebook.com/groups/caherps (accessed on 19 December 2022), *European Herping* https://www.facebook.com/groups/europeanherping (accessed on 7 December 2022), *Field Herping* https://www.facebook.com/groups/fieldherping (accessed on 7 December 2022), *Field Herpetology* https://www.facebook.com/groups/fieldherpetology (accessed on 23 January 2023), *Fieldherping Europe* https://www.facebook.com/groups/251247414973280 (accessed on 6 December 2022), *Field Herping North America* https://www.facebook.com/groups/133777793480466 (accessed on 9 December 2022), *Herping Arizona* https://www.facebook.com/groups/533306516687033 (accessed on 9 December 2022), *Herping Lao* https://www.facebook.com/groups/297572084603938 (accessed on 18 December 2022), *Herping Southeast Asia* https://www.facebook.com/groups/118596601887866 (accessed on 18 December 2022), *Herping in Thailand*
https://www.facebook.com/groups/463127500533775 (accessed on 18 December 2022), and *Kentucky Field Herping* https://www.facebook.com/groups/564583394951731 (accessed on 23 January 2023) on Facebook to gather novel records. No public registration of the above protocol was performed.

### 2.2. Database Construction

The main criterion for inclusion was that the study provided a detailed description of individuals affected with any form of spinal deformity or included a list of affected individuals; therefore, primary case reports and observational studies were considered. The search yielded 143 eligible sources (see Appendix A). Note that after the database was closed, 2 additional records were published and incorporated during the review process, along with a previously overlooked source, bringing the total number of sources used to 146. The database was constructed by extracting the following information: order, superfamily, family, genus, scientific and common names, habitat use strategy (terrestrial, arboreal, aquatic, and semiaquatic), location of observation (country and specific locality), geographic region, and biogeographic realm (when specified). Additionally, SW4 coordinates and elevation were included when available, as well as the exact date of observation and the type of malformation (kyphosis, scoliosis, lordosis, kyphoscoliosis, kypholordosis, lordoscoliosis, and rhoecosis).

On some sporadic occasions, particularly in early 20th-century observations, no specific type of malformation was provided [15,81], or incorrect terminology was used [12,23,82]. In such cases, I reviewed the diagnosis based on photographic evidence. Diagnostic method (observation, radiography, and computerised tomography) was also recorded. When possible, morphometric data, including size (in mm; snout-to-vent length [SVL] or carapace length) and body weight (g), as well as sex, were extracted.

The prevalence of spinal deformities was calculated for records that reported observed cases and the total number of individuals examined from the same population. In cases where multiple individuals with the same type of malformation from the same population were reported, I treated each deformed individual as a separate observation but used their total number to calculate prevalence. Therefore, the prevalence value is the same for these observations. Different types of malformations from the same populations were treated separately when calculating prevalence. If a record reported more than a certain number of individuals examined (e.g., >100 individuals), I used 100 as the total number for calculating prevalence.

The names of all reported species were cross-checked against The Reptile Database [83] to obtain the most recent classification for each species, genus, and family, as well as the taxonomy and number of species within each family. All observations were classified to the species level.

### 2.3. Meta-Analysis

Meta-analyses on proportions, such as prevalence, are commonly conducted, particularly in medicine [84], epidemiology [85], and clinical psychology [86]. In such meta-analyses, each study contributes a specific number of observed cases or successes along with a corresponding total sample size. In this analysis, 54 effect sizes was used, each with an associated total sample size, from 42 studies covering 37 species across 15 families (Appendix A).

Proportional data are often not centred around 0.5 and frequently exhibit significant skewness, deviating from a normal distribution [87]. Using raw proportions as the effect size metric in such cases can result in an underestimation of confidence interval coverage around the weighted average proportion and an overestimation of heterogeneity among observed proportions [88]. Consequently, assuming normality may introduce bias, potentially leading to misleading or invalid inferences [89,90]. To address this skewness, I applied a logit (log odds) transformation, converting the proportions into their natural logarithm using the following equation:$${ES}_{l}=\ln\left(\frac{p}{1-p}\right)$$
where *p* represents the proportion.

The meta-analytic approach assumes that variance between observations due to sampling error can be approximated by the squared standard error. However, sampling variance is higher for low-precision estimates, such as those based on a small sample size or a low number of repeats. This can result in (i) low-precision studies receiving disproportionate weight, (ii) an overestimation of biological variation, and (iii) potential bias in the mean effect size if publication bias is more prevalent among low-precision studies. Sampling variance was calculated using the following equation (following Wang [87]):$${Var}_{l}=\;\frac{1}{np}+\;\frac{1}{n(1-p)}$$
where *p* represents the proportion, and *n* is the total sample size.

Preliminary descriptive analyses revealed a strong association between study identity and species. Except for Bárcenas-Ibarra et al. [32], Rhodin et al. [91], Mitchell et al. [92], and Jackson & Zappalorti [93], all studies focused on a single species, while some species were observed by multiple studies (*Caretta caretta*, N = 2; *Chelonia mydas*, N = 2; *Dermochelys coracia*, N = 2; *Lepidochelys olivacea*; N = 4; *Crysemys picta*; N = 5; see Appendix A). I added species ID to account for sources of non-independence arising from effect sizes coming from the same species. At the same time, effect size ID was added to account for variation in effects across individual effect sizes.

Biological and environmental differences between studies are expected to affect the variation in association between components of behavioural strategy; thus, I accounted for these potential sources of heterogeneity by extracting and examining the effect of two moderator variables. The following variables were extracted from each study: (i) habitat use strategy (aquatic, semiaquatic, terrestrial, or arboreal) and (ii) habitat type (natural, seminatural, or urban [in the case of seven studies, habitat type was not applicable]). Habitat type was determined through visual inspection of the surroundings of the SW4 coordinates using Google Maps https://www.google.com/maps (accessed on 7 August 2025). Urban habitats were defined as city parks, ponds, or riverbanks located within city limits, whereas seminatural habitats included similar features situated in less frequented rural areas or outside city limits but still potentially subject to substantial human influence.

Information on the phylogenetic history of species was obtained based on published phylogenies available through the ‘Open Tree of Life’ [94] via the *rotl* package [95]. Taxon names were matched to records in the Open Tree Taxonomy to obtain relationships between species. It should be noted that, for compatibility with the OTL database, the Suwannee snapping turtle (*Macrochelys suwanniensis*) was treated as a synonym of the alligator snapping turtle (*M. temminckii*). However, this does not significantly affect the interpretation of the results. Due to the diversity of species in this meta-analysis, accurate estimation of branch lengths was not possible; thus, branch lengths were computed based on topology (see Appendix A) using the *ape* package (version 5.8-1) [96] in R (version 4.5.1.) [97]. Phylogenetic heritability or phylogenetic signal (H^2^) was calculated as the proportion of total variance in effect size that can be explained by phylogenetic variance [98], equivalent to Pagel’s λ [99]. H^2^ = 0 indicates no phylogenetic relatedness among effect sizes [100].

I conducted the meta-analysis using a random-effects model, estimated by the restricted maximum likelihood method in the *metafor* package (version 4.8-0) [101]. I first ran an intercept-only mixed model (with random effects) to determine the mean effect size across all studies. To estimate heterogeneity of effect sizes, I used I^2^ statistics [102,103,104] modified for multilevel meta-analytic models [105]: total heterogeneity (I^2^_total_) was partitioned into phylogenetic variance (I^2^_phylogeny_), species ID variance (I^2^_species_), study ID variance (I^2^_study_), and residual variance (I^2^_residual_). Low, moderate, and high heterogeneities refer to I^2^ of 25%, 50%, and 75%, respectively [106]. Next, I constructed a series of meta-regression models to identify the most important moderators (see above) [102]. As the sample size was somewhat limited, I chose to avoid complex models; instead, I conducted fixed-effect mixed models to estimate the mean effect size for each moderator separately [102,107].

Data and detailed code for the analyses can be found at the Open Science Framework (OSF): https://osf.io/agnzt/?view_only=6c5845bde2f54496af577165771ccb6b, accessed on 7 August 2025.

## 3. Results

### 3.1. Overview of Published Cases

In total, 690 observations of spinal deformities were extracted across 24 families, 65 genera, and 109 species (see Appendix A). The records show an increasing tendency over time (Figure 2), with the oldest observation dating back to 1898: a specimen of *Crocodylus acutus* found dead at an undisclosed location in Panama [108]. The overwhelming majority of observations came from the peer-reviewed literature (643 records, 93.2%), including journal articles (576 records, 83.5%). Note, that a single journal article [109] contributed almost half of the records (333 records, 48.3%), and a single dissertation [35] contributed 61 records (8.8%). Books accounted for six records (0.9%). Online social and popular media platforms provided 10 records (1.5%), and the grey literature contributed 37 records (5.36%), all of which were unpublished observations. The study from Sönmez and Sağol [109] places Türkiye as the country with the highest number of records, followed by the USA (157 records, 22.8%), Brazil (65 records, 9.4%; note that without the dissertation of Carvalho [35], Brazil’s share drops significantly to 4 records, or 0.6%), and Mexico (46 records, 6.6%). However, when considering the number of observed species per country, the USA ranks first with 44 species (40.4%), followed by Mexico with 14 species (12.8%), and Spain and Argentina sharing third place with 6 species each (5.5%) (see Figure 3). All observations could be linked to a country, and the specific locality was provided or could be inferred for most of them (503 records, 73%). Exact coordinates were available in only 419 cases (60.7%), and elevation was recorded in 405 cases (58.7%). However, the picture is more limited when examined by source: although locality information was provided in most of them (130 sources, 89%), exact coordinates were available for only 58 (39.7%) and elevation data for just 37 (25.3%).

More than half of the spinal deformities were cases of kyphosis (523 records, 75.8%; in 5 cases, the exact diagnosis was questionable) (Figure 4). Scoliosis was reported in 81 cases (11.7%; 11 cases were questionable), and in one instance, it was not possible to distinguish between scoliosis and kyphosis. Kyphoscoliosis was indicated in 63 cases (9.1%; 4 cases were questionable). Only 1.8% of cases involved lordosis (13 cases, 1 of which was questionable). Rhoecosis was reported in five cases (0.7%), kypholordosis in two cases (0.3%), and there was a single case of lordoscoliosis (0.1%). In most cases, diagnoses were made by visual observation of the specimens. Imaging techniques—such as radiography and computer (micro)tomography—were used to diagnose spinal malformations in 94 cases (13.6% of all observations), reported in 29 studies (20% of all studies). Some form of additional information about the affected specimens was available in many cases. Sex was reported for 202 observations (29.3%), age for 568 observations (82.3%), size (snout–vent length or carapace length) in 167 cases (24.2%), and body mass in 73 cases (10.6%). At the source-level, sex was reported in 87 sources (59.5%), age by 134 (91.7%), size by 79 (54.1%), and body mass by 38 (26%). The origin of spinal deformities (congenital vs. postnatally acquired) was reported or could be extracted for 468 records (67.8%), while possible cause of the observed malformations was suggested by the author(s) in 94 cases (62.3%). Information on whether or not the malformations affected locomotion could be gathered in 68 cases (9.9%).

### 3.2. Prevalence of Spinal Malformations

The distribution of malformation types among the 54 effect sizes closely mirrored that of the full dataset: 33 cases (61.1%) were kyphosis, 10 (18.5%) scoliosis, 8 (14.8%) kyphoscoliosis, and 2 (3.7%) lordosis.

Based on the phylogenetically controlled meta-analysis, the overall mean effect size—calculated as the log odds of proportions—was statistically significant, corresponding to a mean malformation prevalence of 0.21% (this value is the back-transformed estimate from the log odds; see the provided R code for details) (estimate = −6.16, 95% CI = −6.82 to −5.5, t_(df = 53)_ = −18.62, and *p* < 0.001). Importantly, this value represents a weighted average across all included studies or populations, with individual effect sizes in separate populations potentially being higher or lower. Total heterogeneity across effect sizes was high (I^2^_total_ = 89.87%) and was explained by study differences (I^2^_study_ = 41.04%) and effect size differences, representing residual variation (I^2^_residual_ = 41.84%), while both phylogenetic and among species differences were low or negligible (I^2^_phylogeny_ = 6.62%; I^2^_species_ <0.001%). In line with this, heritability was negligible too (mean H^2^ = 0.46).

On average, prevalence for aquatic taxa was 0.19% (estimate = −6.26, 95% CI = −7.32 to −5.22, t_(df = 50)_ = −11.97, and *p* < 0.001), 0.26% in semiaquatic taxa (estimate = −5.93, 95% CI = −7.72 to −4.14, t_(df = 50)_ = −6.64, and *p* < 0.001), 0.26% in terrestrial taxa (estimate = −5.95, 95% CI = −7.61 to −4.31, t_(df = 50)_ = −7.26, and *p* < 0.001), and 0.22% in arboreal taxa (estimate = −6.09, 95% CI = −8.68 to −3.52, t_(df = 50)_ = −4.74, and *p* < 0.001). When comparing habitat use strategies, none of the differences was statistically significant (Figure 5a). On average, prevalence in natural habitats was 0.13% (estimate = −6.66, 95% CI = −7.36 to −5.95, t_(df = 43)_ = −19.08, and *p <* 0.001), 0.21% in seminatural habitats (estimate = −6.16, 95% CI = −7.54 to −4.78, t_(df = 43)_ = −8.99, and *p <* 0.001), and 0.21% in urbanised habitats (estimate = −6.15, 95% CI = −8.32 to −3.98, t_(df = 43)_ = −5.72, and *p <* 0.001). Again, there was no statistical difference between habitat types (Figure 5b).

The funnel plot showed no visual sign of funnel asymmetry (Appendix A), and there was no detectable trend suggesting that more recent publications consistently showed lower or higher effect size values, which would have indicated the presence of time-lag publication bias (estimate = −0.01; 95% CI = −0.03 to 0.01, t_(df = 52)_ = −0.71, and *p* = 0.22; Appendix A).

## 4. Discussion

### 4.1. Affected Reptile Families, Global Distribution, and Reporting

The observations were distributed across 24 families, representing one fourth of the 94 recognised reptile families (as of May 2025 [83]). The proportion of affected families (relative to the total number of families in each group) was highest among crocodilians, at 66.6% (2 out of 3 families), followed by Testudines (57.1%, 8 out of 14), lizards (29.7%, 11 out of 37), and snakes (9.3%, 3 out of 32). No observations were recorded for tuataras or amphisbaenians (see Appendix A). It is perhaps not far-fetched to suggest that the total of 690 records is almost certainly a significant underestimate of vertebral malformations in reptiles. The frequency of recorded cases appears to be closely related to how easily a specific group of reptiles can be observed. For example, the relatively low proportion of affected snake families likely results from the fact that snakes are generally more secretive, solitary predators, often with low or nocturnal activity, whereas lizards and Testudines are more active during the day and typically occur at higher population densities [110]. Fossoriality—a widespread evolutionary strategy among both lizards and snakes—is another factor that may contribute to the lack of observations: No cases of vertebral deformities were recorded from entirely or highly fossorial lizard taxa such as *Anguidae* and *Amphisbaenia*, while the proportion of affected families was extremely low in *Scincidae* (0.22%). In addition to observational biases, the perceived absence of vertebral deformities in fossorial lizards and snakes may also result from the fact that subterranean environments can buffer against certain environmental factors that contribute to malformations, such as climatic fluctuations and pathogens [111].

Another key conclusion that can be drawn from the database is that observations are more common in taxa that are well studied, popular among non-professionals, and exhibit deformities that are easily detectable. Testudines generally fulfil these criteria, most notably because kyphotic and kyphoscoliotic individuals display a characteristic ‘humpbacked’ appearance that is highly conspicuous, even from a distance. This may help explain why Testudines account for 543 records in total (78.7% of all data), the majority of which come from the peer-reviewed literature (514 records, or 74.4%). In contrast, among the novel and unpublished records gathered from Internet sources and personal communications, Testudines represent a slightly lower proportion: 29 records (61.7%). Another factor likely contributing to the overrepresentation of Testudines in the dataset is the prevalence of studies on hatching success in turtle nests [31,32,33,34,35,36,52,109]. Malformed embryos and hatchlings—especially in sea turtles—are frequently reported in large numbers [109], with these records alone contributing for 377 observations (69.4% of all Testudines records). The popularity and well-studied status of turtles among reptiles may explain why this group is not only over-represented, but also why most extant families are affected by vertebral malformations. The proportion of affected species is particularly high in *Dermochelyidae* (100%) and *Cheloniidae* (66.6%)—perhaps the most studied and monitored groups of turtles [112]—but also in *Chelydridae* (60%), and *Emydidae* (46.6%). Interestingly, only three records coming from species that are considered terrestrial (*Terrapene carolina*, *T. ornata*, and *T. triunguis*).

The dataset spans 37 countries (see Figure 3). However, when interpreting geographical trends, I chose to exclude the embryological studies by Carvalho [35] and Sönmez and Sağol [109], as they contribute a large number of records (394, or 57.1%) from only one or two species, potentially introducing significant bias by artificially inflating the geographical importance of their respective countries (i.e., Brazil and Türkiye). With these studies excluded, the dataset is reduced to 292 records, the majority of which originate from the Americas (226 records, 77.4%). Unsurprisingly, the United States provided the largest number of observations (157), mainly due to two factors. First, 140 of these are turtle records, representing 66.6% of all turtle records. This is in line with the result we get when considering the number of observed Testudines species in the reduced dataset (32 records or 69.6%). This aligns with the fact that global turtle species richness peaks in the southeastern United States [113]. Regarding squamates, the United States contributes 16 records, also the highest count in the reduced dataset. Considering the number of observed species, 12 species (21.8%) of lizards and snakes were observed in the United States. This latter pattern does not correspond with global biodiversity hotspots for lizards and snakes but is likely attributable to the large number of professional herpetologists and herpetology enthusiasts in the United States.

Overall, the records are not evenly distributed across the globe and do not reflect global reptile species richness. Although 114 records (39%) originate from the Global South, several biogeographical regions considered hotspots of reptile diversity are significantly under-represented: the Afrotropical region (14 records, 4.8%), Australasia (13 records, 4.4%), and the Indomalayan region (13 records, 4.4%). This is particularly surprising when compared to studies using similar methodologies [114] and suggests that the dataset is biased by differences in language, culture, and scientific practices, factors known to hinder the publication or accessibility of certain records [114,115,116]. It should be noted that although I made every effort to include all available records published in English, German, Spanish, Portuguese, French, and Russian, it is still possible that some non-English publications were missed.

Another factor that may contribute to the uneven global distribution of spinal deformity records is that such observations were—and to some extent still are—regarded as mere curiosities. This likely explains the limited number of peer-reviewed publications, typically no more than one or two per decade until the mid-1900s (Figure 5). Notably, there has been a significant increase in both the number of reported cases and the diversity of affected species since the early 2000s (Figure 5). Despite this growth, the majority of reports are still published in the form of natural history notes. For example, *Herpetological Review* alone accounts for 72 records (29.6% of records coming from journal articles) across 40 species. While most of these publications lack detailed discussion of ecological implications, a few studies have begun to address this dimension [34,72,117,118].

### 4.2. Does Phylogeny and Habitat Use Strategy Affect Prevalence?

Based on the results of the phylogenetic meta-analysis, differences across species (I^2^_species_) were virtually negligible, and phylogeny (I^2^_phylogeny_) appear to be of low significance. This is supported by the estimate of phylogenetic heritability (H^2^), which indicates that only a small proportion of variance can be attributed to phylogenetic relatedness. However, it should be noted that the phylogenetic tree used in the analysis was constructed from a limited number of species and, more importantly, was strictly topological, lacking branch length information. Additionally, the results may be affected by limited statistical power in some taxa; notably, 79.6% of the prevalence estimates—43 records—came from turtles (26 species); therefore, the level of phylogenetic inertia detected in this study should be considered preliminary.

Vertebral malformation prevalence did not differ significantly among (semi)aquatic, terrestrial, and arboreal habitat use strategies (Figure 5a). However, in the dataset, habitat use strategy is highly correlated with phylogeny, as all but one of the (semi)aquatic taxa are turtles. Therefore, this finding complements the previous result on low among-species and phylogenetic variation rather than indicating an effect of adaptation to different environments. Phylogenetic overlap is less pronounced when comparing terrestrial and arboreal taxa; nevertheless, the small sample size imposes substantial limitations.

Another aspect to consider is the distribution of various malformation types across habitat use strategies in the complete dataset. Kyphosis was significantly more frequent in (semi)aquatic taxa (89% of records) than in terrestrial and arboreal taxa (27.8% and 3.7% of records, respectively). Conversely, scoliosis was more frequent in terrestrial and arboreal taxa (28.6% and 59.2% of records, respectively) than in (semi)aquatic taxa (5.85% of records). This pattern aligns with the notion that kyphosis predominantly affects (semi)aquatic turtles, whereas scoliosis is more common in predominantly terrestrial squamates. Nevertheless, this difference is most likely due to substantial differences in the development and structure of the turtle and squamate vertebral column [119,120]. As the database largely lacks records from (semi)aquatic squamates—e.g., water snakes (*Nerodia* and *Natrix*), water monitors (*Varanus* spp.), and semiaquatic agamids such as the Chinese water dragon (*Physignathus cocincinus*) and the Australian water dragon (*Intellagama lesueurii*)—it remains an open question whether the observed differences are due to habitat use strategy (aquatic vs. terrestrial) or rather to phylogenetic, structural, and/or developmental constraints. This issue can only be resolved once more data become available from the aforementioned squamate taxa.

### 4.3. Does Environmental Stress Increase Prevalence?

The prevalence of vertebral deformities did not differ significantly among natural, seminatural, and urbanised habitats (Figure 5b). Urban environments are recognised as multi-stressor landscapes [121], where individuals are exposed to increased chemical pollution, noise and artificial light pollution, infectious diseases, and poor diet quality [122,123] compared to natural habitats. Since several of these environmental stressors have been linked to developmental malformations, the absence of a habitat effect on prevalence may seem surprising. However, it is important to note several limitations of the dataset that may have contributed to this negative result. First, I must reiterate that habitat characterisation was performed based on a predefined set of criteria, and it is possible that some misclassifications occurred during the process. Most records originated from habitats identified as (semi)natural, with only three records coming from urbanised habitats (representing 5.5% of the prevalence data), which limits the statistical power to detect weak effects. The low number of prevalence records from urban habitats is somewhat unexpected, as urban reptile populations tend to be well monitored [124,125,126].

Importantly, from an ecological perspective, stress is not universal: it is linked to the regulatory capacity of an organism [127]. Therefore, an environment that is stressful for one taxon may not necessarily pose stress for another. Reptile taxa that have successfully colonised urban habitats may be less sensitive to the stressors typically associated with these environments. While the urban–seminatural–natural trichotomy can serve as a useful proxy for environmental stress, in the absence of detailed data, it was not possible to evaluate the actual level of environmental stress at individual sites. As a result, important differences may remain obscured. The lack of information on potential environmental stressors and the general condition of each habitat represents a significant limitation of the dataset. Only three publications explicitly stated that the habitat was contaminated [50] or referenced prior human disturbance likely to have degraded the environment [128]. It is essential for future studies and reports to provide information on factors that may act as environmental stressors—such as levels of habitat degradation or contamination—ideally using a standardised framework.

### 4.4. Potential Ecological and Behavioural Effects

#### 4.4.1. Survival and Growth

It is commonly assumed that the low prevalence of spinal deformities in wild populations results from the low survivability associated with such conditions. However, this likely depends on (i) whether the malformation is congenital, (ii) the severity of the malformation, and (iii) whether it forms part of a multifaceted pathology. Information about ontogenetic stage was provided in 569 cases, a substantial number of which were adults or subadults (166 records, 29.2%). Not considering embryological studies from Carvalho [35] and Sönmez and Sağol [109], the share of adults and subadults rises significantly (94.9%). These patterns suggest that in a substantial number of affected individuals, deformations do not necessarily impair survival. Nevertheless, it is questionable in how many of these animals the condition was congenital. This is difficult to determine, as reflected by the fact that a clear congenital origin could be suggested in only five cases, while two was acquired postnatally. Juveniles reflect a similar picture: out of 22 records, in only 6 was possible to tell the origin, half of which was congenital, while the other half acquired postnatally. Regarding embryos, hatchlings, and neonates (453 records), congenital origin is unambiguous. Several studies provide detailed accounts of vertebral malformations affecting embryos, neonates, and hatchlings, often in large numbers within the litter [31,32,34,35,36,52,53,109,129]. In most cases, spinal deformities were accompanied with other lethal anomalies incompatible with life. Some studies report that affected juveniles died at a very young age [130,131], whereas others found no observable negative effects of vertebral malformations on hatchling, neonates, or juveniles [130,132]. If most spinal deformities are indeed congenital, these patterns suggest that malformations affect a large number of embryos, of which only a few survive. This is supported by studies on sea turtles, where examinations typically involve a large number of embryos and hatchlings [53,109]. It is also likely that many malformed individuals die at a young age. Nevertheless, once an individual reaches adult size, vertebral malformations do not necessarily impair survival. In fact, many reports explicitly state that affected juveniles and (sub)adults were in relatively good condition and health [9,68,133,134,135,136,137,138,139,140,141], sometimes despite the severity of malformation [142].

Regarding long-term survival and growth patterns, data are available exclusively from long-lived turtle taxa. Information provided by Inwater Research Group Inc. includes kyphotic loggerhead sea turtle (*Caretta caretta*) and green sea turtle (*Chelonia mydas*) specimens that were recaptured multiple times over a decade. Moldowan et al. [132] reported a kyphotic painted turtle (*Chrysemys picta marginata*) that evidently lived for at least 17 years, with its condition apparently having no effect on its growth or fitness. Mitchell and Johnston [72] reported a Florida chicken turtle (*Deirochelys reticularia chrysea*) with kyphoscoliosis that was recaptured three years after the initial encounter and exhibited a seemingly normal growth pattern. Conversely, Harding and Bloomer [143] and Selman and Jones [144] found that kyphotic wood turtles (*Glyptemys insculpta*) and ringed map turtles (*Graptemys oculifera*) grew more slowly than expected. Based on these findings, we may conclude that—at least in turtles—vertebral deformities do not hinder normal growth or have only mild effects on it.

#### 4.4.2. Escape and Predation

Besides direct negative effects on viability, vertebral malformations may increase mortality by making affected individuals more susceptible to predation, mainly due to limited escape capabilities. In the dataset, there were 72 records where information on locomotor abilities was provided, but only 13 studies (18.1%) reported any clear locomotor disorders. These cases involved various, usually severe, types of malformations, suggesting that relatively mild vertebral deformities do not necessarily hinder locomotion, regardless of their type. For example, Mitchell and Georgel [135] reported a juvenile kyphoscoliotic Eastern fence lizard (*Sceloporus undulatus undulatus*) that actively pursued prey in captivity. Hernández et al. [145] described a Mesquite lizard (*Sceloporus grammicus*) with thoracic kyphosis and pelvic scoliosis that was observed multiple times moving with speed and agility among rocks. Several malformed turtles were also found to move and swim normally [37,141]. In comparison, Avellá Machado and Acosta [146] mentioned a Darwin’s tree iguana (*Liolaemus darwinii*) that ran without difficulty, although its movement appeared more rigid than that of a healthy individual.

Nevertheless, even if vertebral deformities do not substantially affect locomotor performance, they may compromise the effectiveness of certain antipredator behaviours. For instance, Elsey et al. [142] reported a severely malformed red-eared slider (*Trachemys scripta elegans*) that appeared healthy and mobile, but its condition prevented complete forelimb retraction and caused abnormal head retraction. Similarly, Hirst et al. [147] reported a kyphotic spiny chuckwalla (*Sauromalus hispidus*) that seemed to have a limited ability to effectively hide in a burrow. Such limitations are mostly associated with kyphosis and kyphoscoliosis and may particularly affect taxa that rely on crevices and burrows for shelter.

A special type of vertebral deformity also warrants mention in this context: caudal scoliosis, where the malformation is restricted to the tail region, is generally considered unlikely to have severe negative effects on any aspect of the specimen’s life. Records of caudal scoliosis are known only from squamates and accounted for 10% of all 140 squamate observations in the database. Although none of these reports mentioned clear locomotor disorders or disadvantages, tail morphology has been shown to influence locomotor performance in lizards [148], though not in snakes [149]. Therefore, we may expect some differences between individuals with normal and curled tails, but this issue can only be resolved through targeted measurements.

#### 4.4.3. Reproductive Outcome

Spinal deformities—especially severe ones—are expected to affect reproductive outcome on multiple levels. First, such malformations may hinder mating success, primarily by impairing courtship and mating behaviours or by making copulation physically impossible. In addition, fertility measures, as well as the number and viability of offspring, may be reduced in malformed mothers. To date, only sporadic explicit or indirect information on the potential effects of spinal deformities on reproductive outcome can be found in the literature. Importantly, the database indicates that both sexes are affected by spinal deformities, although female records are nearly twice as numerous as male records (124 vs. 77; 61.7% vs. 38.3%). However, since sex was not reported in most cases (489, or 70.9% of all records), these proportions may not accurately reflect the true sex ratio.

Although it does not count as mating behaviour *per se*, Norval et al. [150] mention an adult male scoliotic japalure (*Diploderma chapaense*) that was engaged in a territorial dispute when captured. This observation is important because it represents the only published case providing information on an individual with vertebral deformity expressing specific social behaviour in the wild. Despite not knowing the outcome of this agonistic encounter, it seems not too far-fetched to say that behaviours linked to mating and courtship are expected to be affected alongside locomotor performance, especially in lizards, where active chasing and grabbing of females by males is very common [151,152]. As we have seen, cases with clear indications of locomotor dysfunction, although substantial in number, represent only a minority of records; thus, we may rightly deduce that mating behaviour is relatively rarely affected as well. Nevertheless, beyond actively pursuing their mate, both males and females of several species—especially iguanids—are known to exhibit quite elaborate displays during mating [153,154], including swinging or curling the tail, behaviours that are surely affected even by mild malformations such as caudal scoliosis. No first-hand information regarding this is available, however.

Regarding copulation success of affected males, we possess no first-hand knowledge. While severe malformations may hinder males across various taxa, in turtles—where kyphosis is seemingly more common—females might be more likely affected, even by relatively mild kyphotic malformations. There is a sole observation that indicates otherwise or at least suggests that vertebral malformations do not necessarily pose a problem in turtles when it comes to copulation. Burke [155] reports an adult female Eastern spiny softshell turtle (*Apalone spinifera*) with extreme kyphosis that was gravid and laid a clutch of eggs a few weeks after its capture (oviposition was induced by oxytocin injection). Sadly, the eggs were laid in water, so it was not possible to determine the exact number or fertility of the eggs.

Owens and Knapp [156] provide valuable information on the number and viability of offspring. They describe a kyphoscoliotic female Andros Island iguana (*Cyclura cyclura cyclura*) that nested in a termite mound. Despite aggressively defending its nest, the nest was excavated, and the eggs inspected. The clutch size of seven eggs was within the normal range for the species; however, the mean egg size was slightly smaller than normal. Despite this, all eggs hatched successfully, and the hatchlings were viable and non-malformed. Owens & Knapp suggest that the spinal curvatures may have resulted in a restricted cloacal opening and constrained egg width, while a reduced abdominal cavity may have caused the below-average egg mass. The existence of such additional effects and their distribution across various taxa could be relatively easily confirmed using modern imaging techniques on malformed females.

## 5. Conclusions and Future Directions

Reports of spinal deformities in wild reptiles were sporadic throughout the last century. However, over the past 25 years, notes and short communications documenting individuals with such malformations have become increasingly common, with nearly ten publications per year on average. This rise has enabled the production of some thematic syntheses [91,117], although these studies offer limited insight into ecological implications. Most publications remain restricted to single-case observations that include morphological measurements and, occasionally, data on occurrence frequency. Another issue is that despite the growing number of peer-reviewed papers, spinal deformities are still often regarded as mere curiosities, even by professionals. As a result, a substantial number of observations likely remain unpublished or are available only in the grey literature, where they tend to go unnoticed.

This review presents the most comprehensive documentation and discussion of spinal deformities—primarily kyphosis, scoliosis, and kyphoscoliosis—in wild reptiles to date. It is safe to conclude that published reports significantly under-represent the true number of spinal deformity cases in wild reptiles. Observation frequency appears to be more strongly influenced by research focus and taxonomic popularity than by actual incidence, while the geographical distribution of reports likely reflects the presence of professional herpetologists and amateur enthusiasts rather than underlying species richness. Meta-analytic results revealed no significant effect of phylogeny, habitat use strategy, or habitat type. However, the absence of detectable effects is likely due to limitations in the available data. Notably, the significant over-representation of (semi)aquatic turtles may contribute to low phylogenetic variation. Additional data from squamate taxa—particularly those with (semi)aquatic lifestyles—are needed to better assess potential phylogenetic differences.

Spinal deformities have the potential to impact individual ecology and life history traits, particularly escape and survival, growth, and reproductive output. However, direct data are scarce, and current inferences can only be drawn from isolated, specific cases. Informed by the limited known effects of kyphosis and scoliosis on ecology and life history, this work is intended to spark a lively discussion on the potential ecological implications of spinal deformities. In addition, the present work aims to provide a blueprint for simple approaches to future observations in order to enhance their ecological relevance and our overall understanding of the ecological effects of spinal deformities in wild reptiles.

(i) Observations of spinal deformities—even single cases—are not mere curiosities; they provide important scientific information. Therefore, it is essential to publish them on scientific platforms, preferably in peer-reviewed herpetological journals. In many cases, malformed animals—especially turtles—are observed from a distance and cannot be captured; in such instances, photographic documentation of the specimen(s) is the minimum expected standard.

(ii) Information on the frequency of occurrences within populations and habitat characteristics should be provided. These data can help determine how common spinal deformities are in a given population and identify potential environmental factors—particularly environmental pollution—that may influence their occurrence. In the absence of first-hand information on abundance and population density (which are needed to calculate prevalence), future studies should consider alternative approaches to obtain these data, such as contacting conservation agencies or national parks. Alternatively, online community science data repositories—such as iNaturalist—may provide information on local population abundance. Regarding environmental pollution and habitat degradation, as much information as possible should be collected to identify the most significant environmental stressors. However, if no specific data are available, the level of urbanisation may serve as a useful—albeit limited—proxy.

(iii) It is important to provide information on morphology (snout–vent length and tail length for squamates; carapace length, carapace width, and carapace height for turtles), body mass, ontogenetic stage, and sex.

(iv) The exact type of malformation should be confirmed, along with a detailed description of the affected regions of the spine, the number of curvatures, and any additional abnormalities. Since visual observation is not always sufficient, the use of modern imaging techniques—such as radiography and CT scanning—is strongly encouraged. These methods are valuable for detecting concurrent physical abnormalities that may influence life history traits.

(v) The collection of blood samples is encouraged. Blood cultures offer a relatively simple and minimally invasive method for diagnosing active bacterial osteomyelitis and osteoarthritis. This approach may help advance our understanding of the aetiology of vertebral malformations in wild reptiles.

(vi) Testing potential behavioural, locomotor, and foraging efficiency limitations associated with spinal deformities is essential for assessing the extent to which these malformations are detrimental to the individual. Such investigations can clarify their role in shaping behavioural ecology and provide insight into how spinal deformities affect the survival of various reptile taxa.

## Figures and Tables

**Figure 1 biology-14-01119-f001:**
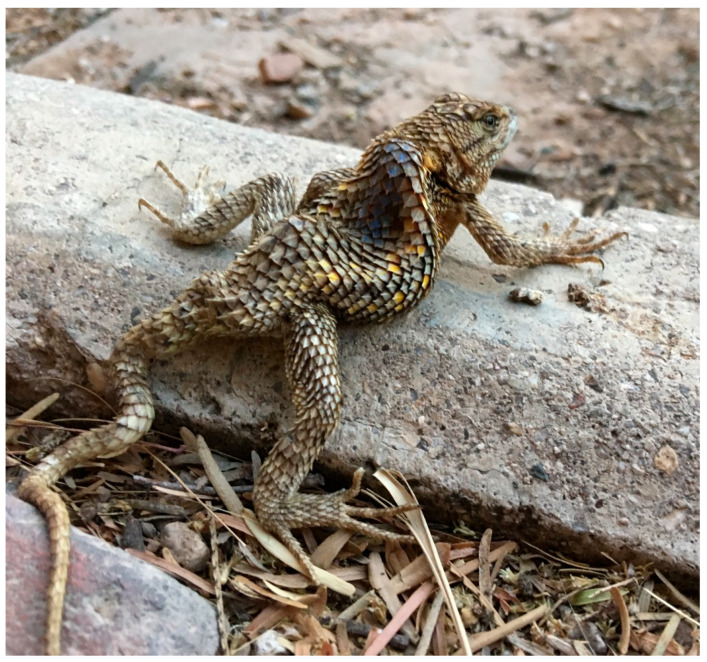
Kyphoscoliotic desert spiny lizard (*Sceloporus magister*). Photograph by Dennis Caldwell, used with permission.

**Figure 2 biology-14-01119-f002:**
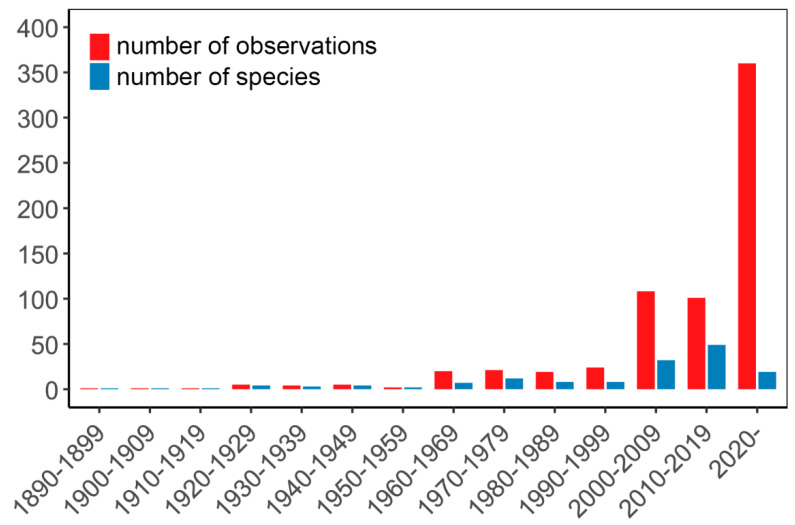
Number of vertebral malformation observations in wild reptiles and the number of affected species according to peer-reviewed and ‘grey’ literature publication records according to the year of observation.

**Figure 3 biology-14-01119-f003:**
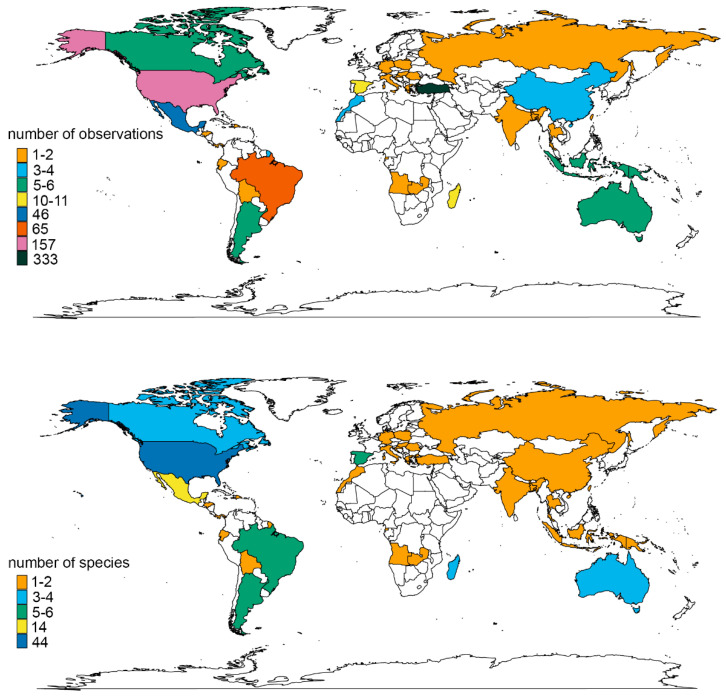
Global distribution of spinal deformity observations in wild reptiles, presented by observed cases (top panel) and observed species (bottom panel).

**Figure 4 biology-14-01119-f004:**
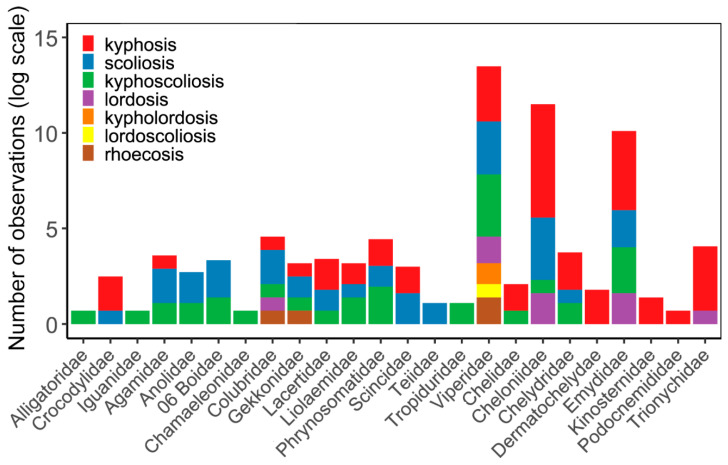
Types of vertebral malformation observations in different reptile families according to peer-reviewed and ‘grey’ literature publication records.

**Figure 5 biology-14-01119-f005:**
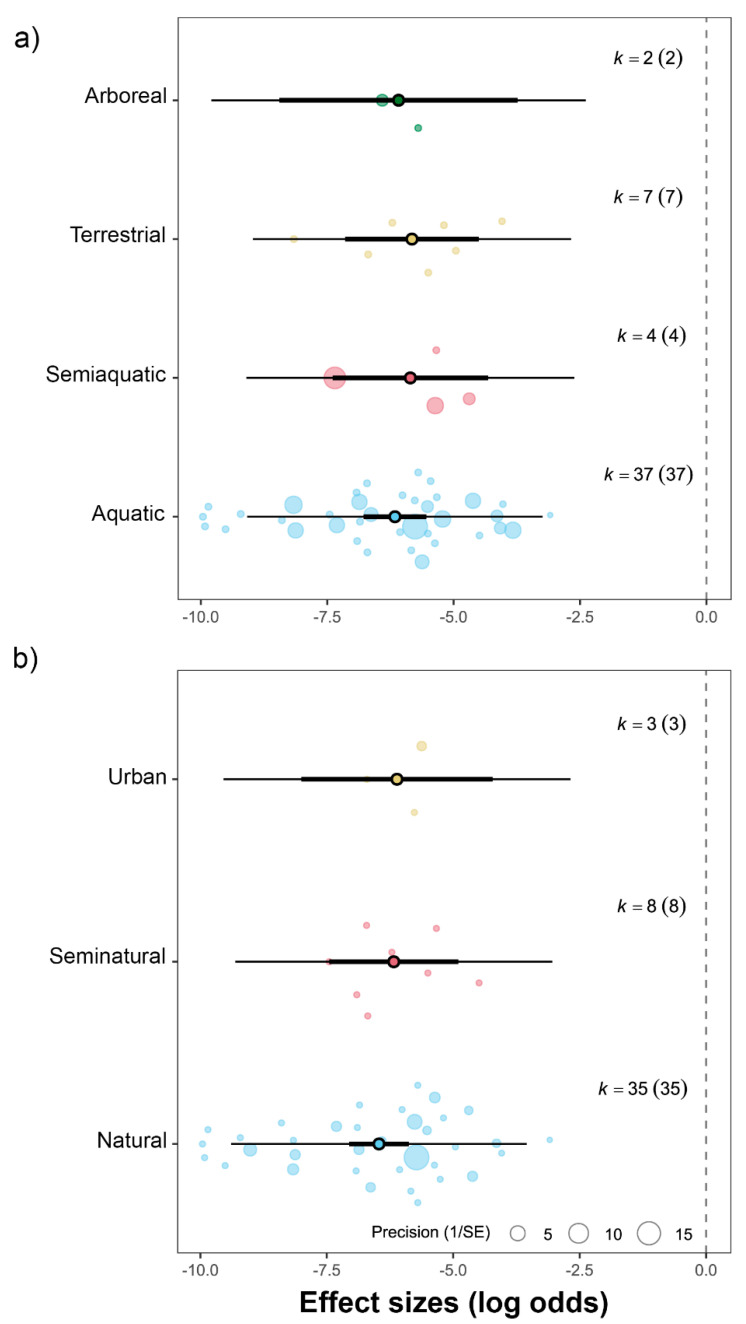
Mean effect sizes by (**a**) habitat use strategies and (**b**) habitat types. Thick horizontal lines represent 95% confidence intervals; thin horizontal lines represent 95% prediction intervals. The point at the centre of each thick line indicates the average effect size. k denotes the number of effect sizes used to calculate the statistics, with the number of studies given in parentheses.

## Data Availability

Data and detailed code for the analyses can be found at the Open Science Framework (OSF): https://osf.io/agnzt/?view_only=6c5845bde2f54496af577165771ccb6b (accessed on 23 June 2025).

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
