# Peer review of "Spinal Deformities in Wild Reptiles: A Systematic Review and Meta-Analysis"

_biology, 2025, doi:10.3390/biology14091119_

Round 1
Reviewer 1 Report
Comments and Suggestions for Authors
Dear Author,
Thank you very much for this comprehensive work. I have highlighted only a few points in the PDF file that I would like to draw attention to in the MS. There has been an extensive literature review and it is really admirable. However, I have a few suggestions for articles and a few minor points that need clarification from you.
Furthermore, to generalize the outputs of the paper, it would be good to include a “Conclusions” subtitle that briefly summarizes the main results.
Best

Author Response
Reviewer #1
Dear Author,
Thank you very much for this comprehensive work. I have highlighted only a few points in the PDF file that I would like to draw attention to in the MS. There has been an extensive literature review and it is really admirable. However, I have a few suggestions for articles and a few minor points that need clarification from you.
Furthermore, to generalize the outputs of the paper, it would be good to include a “Conclusions” subtitle that briefly summarizes the main results.
A: Thank you very much for your positive opinion! Please find my detailed answers to your comments below.
- L 124-125: Not to forget epigenetic mechanisms. Please also read the relevant articles below. https://onlinelibrary.wiley.com/doi/full/10.1111/eva.70013; https://cdnsciencepub.com/doi/abs/10.1139/cjz-2015-0248
A: Thank you for this important remark. I added a sentence to acknowledge the potential role of epigenetic mechanisms based on the recommended literature (L 124-138).
- L 400: There are also records from the Mediterranean (Turkey). Please read the related article below. https://onlinelibrary.wiley.com/doi/full/10.1002/jez.2851
A: Thank you very much for sharing this important, previously overlooked study. Since it contains valuable records, I have not only included it in the reference list but also updated the database. In fact, two additional studies were published after the database was closed in January 2025, and these have now been incorporated as well. The dataset now includes 690 records across 24 families, 65 genera, and 109 species, based on 146 sources.
As a results, the analyses, results and discussion have been modified and updated accordingly (see e.g., L 289-291; Sections 3.1 and 3.2; L 518-549; L 594-597). While the main results remain consistent, the increased number of records and effect sizes made the analyses more robust.
- Figure 2. What do the numbers on the map represent, the number of individuals? Number of studies? Please explain in the figure caption.
A: I have updated this figure with the requested information. Please note that it is now labelled as Figure 3, which shows the global distribution of spinal deformation observations in wild reptiles, presented by observed cases (top panel) and observed species (bottom panel) (L 425-426).
- L 634-636: I suggest you read the following researches for this suggestion. You can also find information about the presence and frequency of Kyphosis and Scoliosis in turtles.
https://onlinelibrary.wiley.com/doi/full/10.1002/jez.2851
https://www.mdpi.com/2076-2615/11/2/444
A: I have added the suggested references to the text and made slight revisions to the passage covering this topic (L 661-664).
- L 642-645: When you say recaptured, what do you mean, individuals with malformations are recaptured? Please give more explanatory information. This sentence sounds like a gloss and does not give clear information.
A: The recaptured individuals were kyphotic. I clarify this now in the text (L 671).
- I think it would be better to write a paragraph of conclusions. It would be good to briefly outline the main conclusions in order to generalize the outputs of this article.
A: The main conclusions are now included in the final paragraph, which has been renamed as “Conclusion and future directions” (L 780 – 795).
Reviewer 2 Report
Comments and Suggestions for Authors
This ms. represents a novel approach to gathering insights into skeletal malformations in reptiles. The author is aware of the many shortcomings of this approach (it represents an underestimation, there is observation bias, common species are reported more commonly, and uneven geographic representation, for example) as detailed in the Discussion, yet runs with them a bit too far. I prefer a lighter touch when considering the following: factors causing these malformations, their ecological importance, global distribution, prevalence, phyletic position, habitat effects, stress effects, survivability, ability to avoid predation, and reproductive success. These are areas where the author makes himself vulnerable to refutation by future studies.
If I were the author I would ask myself "With the dataset what can I know for certain?" and present these as facts. I would then group all speculation into another, well referenced, section that makes it clear that it is speculation. I would make this section long enough to cover the topic, but short enough that it doesn't dominate the paper, like it does in the Introduction and the last 2/3 of the Discussion.
Some hard-earned insights:
Malformations are structural/functional abnormalities that at least in the short term allow animals to live. For example, frog tadpoles can tolerate limb abnormalities because they do not yet walk. Malformations are not the worst thing that can happen to an animal. I always found it curious when, for example, contamination in a wetland would kill 75% of the tadpoles and cause malformations in the remaining 25% and the thing of interest was the 25% malformations, not the mortality.
Axial skeletal malformations might not be the worse malformations an animal could have. Soft tissue irregularities - failure of ectoderm to fuse correctly with endoderm and cause gut disfunction - can have more serious survival implications. So would limb abnormalities. This should be addressed.
A comparative approach is always useful. Books have been written on amphibian malformations and they give percentages of malformation types.
A malformation incidence of 1% or less is a curiosity and has no real ecological or demographic consequences, a malformation rate of 10 or even 60%, as we have seen in amphibians, can lead to the collapse of a population (but almost never does). When you cover implications of malformations, understand that rate matters.
Bottom line: You cover only axial malformations, not limb or soft tissue malformations, yet speculate wildly on implications. By doing so you open yourself up to all sorts of criticism, and none of us need that. Understand what your dataset says and doesn't say, and keep the speculation understood as speculation and to a minimum.
Specific comments:
We all want to create catchy titles that generate interest but your use of hunchbacks is too narrow and could be seen by the more delicate of scientists as offensive.
The argument I have made is spinal malformations (especially rostral ones) in otherwise normal animals are less likely caused by trauma than developmental glitches/nutritional deficits, because trauma would likely injure the spinal cord, which would lead to observable paralysis or paresis. Spine irregularities in humans are common and almost never the result of trauma.
Your Ecological Implications section (1.2) is weak. The loss of one animal in a population due to a malformation only affects that population if it's on its way out and numbers are low. I suggest considering a set of hypotheses, perhaps under your Future Directions section (5) along the lines of: Due to the ability of water to accumulate and concentrate toxins, aquatic reptiles should exhibit higher rate of malformations; or, Reptiles downwind of Los Alamos should exhibit ...
Author Response
Reviewer #2
This ms. represents a novel approach to gathering insights into skeletal malformations in reptiles. The author is aware of the many shortcomings of this approach (it represents an underestimation, there is observation bias, common species are reported more commonly, and uneven geographic representation, for example) as detailed in the Discussion, yet runs with them a bit too far. I prefer a lighter touch when considering the following: factors causing these malformations, their ecological importance, global distribution, prevalence, phyletic position, habitat effects, stress effects, survivability, ability to avoid predation, and reproductive success. These are areas where the author makes himself vulnerable to refutation by future studies.
A: Thank you for your time you dedicated to reviewing my work! Please find my detailed responses to your comments below. While I agree that the results are limited by various and often substantial bias, I believe that this is inherent by the nature of the data, rather than the approach (i.e., phylogenetic meta-analysis based on a systematic literature review). This is especially true for the uneven geographic distribution and the higher incidence of common, well-known species and clades. These are realities we must contend with when conducting statistical tests on such dataset.
Regarding your comment about potentially overinterpreting the results, with all due respect, I am not entirely sure what you mean by it. The first part of the discussion (i.e., affected families and geographic distribution [Section 4.1], phylogenetic inertia and prevalence across habitat use strategies and habitat types [Sections 4.2 and 4.3]) directly derives from the meta-analytic results and what can be inferred from the published literature. While some patterns can be identified, a degree of speculation is inevitable (After all, isn’t speculation a fundamental part of scientific work?). Nevertheless, I now list the main conclusions at the end of the text in Section 5, where I explicitly emphasize the limitations of my methods and the data (see L 785-788, L 793-795).
If I were the author I would ask myself "With the dataset what can I know for certain?" and present these as facts. I would then group all speculation into another, well referenced, section that makes it clear that it is speculation. I would make this section long enough to cover the topic, but short enough that it doesn't dominate the paper, like it does in the Introduction and the last 2/3 of the Discussion.
A: Please see my previous response, particularly regarding your comments on the Discussion. While I agree that some sections—especially section 4.4—do contain speculation, I believe this is grounded in the available literature. Throughout the text, I have consistently highlighted the limitations and biases of the dataset.
As for the Introduction, it primarily reflects what we currently know about the most common causes of spinal deformations. The challenge, however, is that it is difficult to pinpoint a single cause in most cases as the aetiologies are typically multifactorial. Nevertheless, providing a comprehensive list of the key factors is an important aspect of the review. Given that this section serves as background information, I do not believe it is overly speculative.
Some hard-earned insights:
Malformations are structural/functional abnormalities that at least in the short term allow animals to live. For example, frog tadpoles can tolerate limb abnormalities because they do not yet walk. Malformations are not the worst thing that can happen to an animal. I always found it curious when, for example, contamination in a wetland would kill 75% of the tadpoles and cause malformations in the remaining 25% and the thing of interest was the 25% malformations, not the mortality.
A: I agree that most spinal deformations are not lethal, and I have stated this in the text multiple times even within the same section (see L 649-650, L 660, L 666, L 670-682). However, the primary focus of this work is not on the lethality of spinal deformities, but rather on their potential effects— whether or not they influence individual behaviour and life history, and, in turn, potentially affect the population. Currently, we do not have definitive answers on this, and the present results are not conclusive. Therefore, the main goal of the review is ‘to spark a lively discussion on the potential ecological implications of spinal deformities. In addition, the present work aims to provide a blueprint for simple approaches to future observations, in order to enhance their ecological relevance and our overall understanding of the ecological effects of spinal deformities in wild reptiles.’ (see L 797-800).
Axial skeletal malformations might not be the worse malformations an animal could have. Soft tissue irregularities - failure of ectoderm to fuse correctly with endoderm and cause gut disfunction - can have more serious survival implications. So would limb abnormalities. This should be addressed.
A: Please see my previous response. First, I did not collect data on any other type of malformations and as such I would prefer not to discuss their potential effects. This would not only be more speculative but would also divert the focus from the main topic. Additionally, I am sceptical that malformations of such diverse origin could be directly compared in a meaningful way.
Second, the primary message of the work is not about the lethality of spinal deformations. While I would argue that only (near-)lethal malformations could have significant ecological effects, even if it turns out that spinal deformations do not pose serious ecological consequences, this finding could still be of utmost importance. A relevant example is a recent study from Stroud et al. (Am Nat https://doi.org/10.1086/737525) which examines the effects of limb deficiencies in lizards. The study found no significant differences in general health or locomotor performance between healthy and limb-deficient individuals, which has important implications for natural selection. This work concludes that natural history observations can serve as a catalyst for new conceptual perspectives.
While there are currently insufficient comparative tests regarding reptiles with healthy versus deformed spines, I hope that the work presented here will spark new discussions and ideas for future research on this topic.
A comparative approach is always useful. Books have been written on amphibian malformations and they give percentages of malformation types.
A: While it is possible to compare reptiles and amphibians, such a comparison would need to account for several key differences in life history, developmental traits, and ecological factors that could significantly influence the prevalence, detection, and consequences of spinal deformities in these groups. One of the most important factors is that the aquatic lifestyle and fragile skin of amphibians make them highly sensitive to various teratogens. As the reviewer mentioned earlier, amphibians are also more prone to population-level wipeouts than reptiles. Mortality is particularly high among tadpoles, and severe malformations often lead to early death, meaning these individuals would not be observed. Another critical difference is that, due to the high early mortality in amphibians, spinal malformations are mostly detected in tadpoles, whereas in reptiles, most observations are made in adults (with the exception of sea turtles).
Given these significant differences, I would prefer to exclude amphibians from the current narrative to maintain coherence. Initially, during the early stages of database construction, I did include amphibian data. However, as these differences became clearer, I decided to focus exclusively on reptiles.
A malformation incidence of 1% or less is a curiosity and has no real ecological or demographic consequences, a malformation rate of 10 or even 60%, as we have seen in amphibians, can lead to the collapse of a population (but almost never does). When you cover implications of malformations, understand that rate matters.
A: The global prevalence estimate of 0.21% is essentially a weighted average across all included studies or populations, so individual effect sizes (i.e., prevalence estimates) may vary. This information has now been added to the text (L 461-463). For example, according to Honarvar et al. (2011), the prevalence of kyphosis in the leatherback turtle population at Playa Maoba (Bioko Island, Equatorial Guinea) is 4.35%. Some may consider this is a relatively low frequency, and I agree. However, I want to clarify that I did not suggest anywhere in the text that spinal deformities threaten population decline in reptiles. Nonetheless, they may still have important ecological and even evolutionary implications. Even if it turns out that spinal deformations have no serious ecological effects, such a finding could still be of utmost importance (see Stroud et al., Am Nat, https://doi.org/10.1086/737525).
Bottom line: You cover only axial malformations, not limb or soft tissue malformations, yet speculate wildly on implications. By doing so you open yourself up to all sorts of criticism, and none of us need that. Understand what your dataset says and doesn't say, and keep the speculation understood as speculation and to a minimum.
A: Without data on other types of malformations, I prefer to avoid incorporating them into the narrative to maintain focus and coherence within the study. Regarding the speculative aspects, please see my previous answers. With all due respect, I would argue that my speculations are without basis. The first part of the discussion (i.e., affected families and geographical distribution [Section 4.1], phylogenetic inertia and prevalence across habitat use strategies and habitat types [Sections 4.2 and 4.3]) directly derives from the meta-analytic results and the insights drawn from the published literature. While some patterns can be identified, a degree of speculation is inevitable (After all, isn’t speculation a fundamental part of scientific work?). I now summarise the main conclusions at the end of the text in Section 5, explicitly highlighting the limitations on both the methods and the data (see L 785-788, L 793-795).
Specific comments:
We all want to create catchy titles that generate interest but your use of hunchbacks is too narrow and could be seen by the more delicate of scientists as offensive.
A: I changed the title to ‘Spinal deformities in wild reptiles: A systematic review and meta-analysis’ to avoid any unintended pejorative or inconsiderate tone.
The argument I have made is spinal malformations (especially rostral ones) in otherwise normal animals are less likely caused by trauma than developmental glitches/nutritional deficits, because trauma would likely injure the spinal cord, which would lead to observable paralysis or paresis. Spine irregularities in humans are common and almost never the result of trauma.
A: With all due respect, I am not entirely clear on your point here. At no point in the text do I imply that physical trauma is a direct cause of spinal deformities. I do mention that it was previously considered one of the primary causes (line 84), but in the same sentence, I also state that this is unlikely in reptiles based on the available literature.
Your Ecological Implications section (1.2) is weak. The loss of one animal in a population due to a malformation only affects that population if it's on its way out and numbers are low. I suggest considering a set of hypotheses, perhaps under your Future Directions section (5) along the lines of: Due to the ability of water to accumulate and concentrate toxins, aquatic reptiles should exhibit higher rate of malformations; or, Reptiles downwind of Los Alamos should exhibit ...
A: I agree that Section 1.2 was somewhat weak on ecological implications, the most important of which are addressed later in section 4.4. As such, I have reworded and renamed Section 1.2 as ‘Aims and goals’ as that more accurately reflects the content of this this section.
Regarding the hypotheses you mention, I would prefer not to formulate any at this stage. The meta-regression results do not indicate any effect of habitat use strategy, i.e., prevalence is not higher in (semi)aquatic compared to terrestrial or arboreal ones. At this point, I cannot draw any clear conclusions regarding the role of environmental stress or pollution as key factors driving the emergence of spinal deformities in reptiles. Nevertheless, I hope that this review will spark new research ideas or the emergence of yet unpublished data in the topic that may may eventually help answer these important questions.
Round 2
Reviewer 2 Report
Comments and Suggestions for Authors
This will be a well-cited paper.
Author Response
Reviewer #2
This will be a well-cited paper.
A: Thank you very much again for the time you dedicated to reviewing my work and your positive opinion!